# Preferences for on-demand/intermittent/event-driven and long-acting injectable (LAI) HIV pre-exposure prophylaxis (PrEP) among HIV-negative Black gay, bisexual, and other sexual minority men in the United States: A qualitative study

Adedotun Ogunbajo◉*, Alexa Euceda, Raven Ekundayo, Jamil Smith, Temitope Oke, DeMarc Hickson

Us Helping Us, People into Living Inc., Washington D.C., District of Columbia, United States of America

* dotunogunbajo@gmail.com

## Abstract

### Background

Black sexual minority men (SMM) are disproportionately affected by HIV. Pre-exposure prophylaxis (PrEP) is a medication that reduces HIV acquisition. There is a gap in our understanding of the acceptability of and preference for on-demand and long-acting injectable (LAI)-PrEP among PrEP-eligible Black SMM. This study aimed to explore preference for on-demand and LAI-PrEP and reasons for those preferences among HIV-negative Black SMM of different PrEP use profiles (current PrEP users, current non-PrEP users, and PrEP discontinuers).

### Methods

Between March 2022 and April 2023, we conducted 17 focus group discussions (FGDs) with a total of 58 HIV-negative (current PrEP users, non-PrEP users, and PrEP discontinuers) Black SMM residing in the Washington D.C. metropolitan area. We explored acceptability and interest in on-demand and LAI-PrEP and anticipated barriers and facilitators to uptake.

### Results

Two key themes emerged around interest and preferences for on-demand PrEP and LAI-PrEP: 1) lack of interest in on-demand PrEP, and 2) high acceptability of LAI-PrEP. The reasons for lack of interest in on-demand PrEP were: 1) inability to accurately anticipate and plan for sexual activity in advance, 2) uncertainty about effectiveness of on-demand PrEP, and 3) potential for unnecessary medication use, especially when anticipated sexual activity doesn't occur. Most participants finding

**Data availability statement:** The data is publicly available and can be found at:https://data.qdr.syr.edu/dataset.xhtml?persistentId= doi:https://doi.org/10.5064/F6AW0DHS DOI: https://doi.org/10.5064/F6AW0DHS

**Funding:** This study was funded by the National Institutes of Health (R34DA054870). Dr. Ogunbajo acknowledges salary support from U.S. National Institutes of Health K01MH129165.

**Competing interests:** The authors have declared that no competing interests exist.

LAI-PrEP to be highly acceptable can be attributable to: 1) LAI-PrEP being convenient, and 2) LAI-PrEP being a potential solution to suboptimal adherence to daily oral PrEP due to forgetfulness.

## Discussion

While daily oral PrEP is the most utilized PrEP modality, some SMM—who might not be acceptable to oral PrEP—are capable of accurately identifying specific periods and circumstances for HIV infection. It is important that healthcare providers present on-demand PrEP as an option to individuals who fall into these categories. Programs to increase awareness and knowledge of LAI-PrEP among Black SMM and HCP that serve them are needed. Additionally, it is imperative that HCP who provide PrEP services receive proper training on the administration of LAI-PrEP and that there is dedicated staff to help clients navigate the insurance coverage process.

## Introduction

Black people in the United States (U.S.) have a higher proportion of new HIV diagnoses and people living with HIV (PLHIV) compared to any other racial or ethnic group [1,2]. In 2019, sexual minority men (SMM) including gay and bisexual men, accounted for most (70%) of all new HIV diagnoses in the U.S. Black SMM are disproportionately affected by HIV, accounting for 26% of all new HIV diagnoses and 37% of new diagnoses among all SMM. HIV pre-exposure prophylaxis (PrEP) is a highly effective preventative medicine that reduces the potential chance of contracting HIV through sex by 99% among SMM [3]. However, PrEP uptake among Black SMM remain suboptimal. The Centers for Disease Control and Prevention (CDC) estimates that only 13% of Black people who could benefit from PrEP are currently prescribed it [4]. Additionally, a national study of SMM found that Black SMM were less likely than white SMM to report PrEP awareness and use and less likely = to follow a provider's recommendation to initiate PrEP [5]. Barriers to PrEP uptake among Black SMM include potential costs, fear of side effects, HIV-related stigma, racial discrimination, substance use, negative experiences with healthcare providers, and mistrust of the healthcare system [6–8]. Increasing PrEP utilization and adherence especially among Black SMM is imperative to reducing HIV incidence and achieving a 90% reduction in new HIV infections by 2030.

In July 2012, the U.S. Food and Drug Administration (FDA) approved the first medication to be utilized as PrEP for HIV prevention [9]. Since that approval, the most common modality for PrEP usage has been a daily oral pill. More recently, new PrEP modalities, including on-demand and long-acting injectables (LAI), have emerged. In December 2021, the CDC updated its PrEP clinical practice guidelines to include information about the use of off-label 2-1-1 dosing for oral PrEP, often referred to as on-demand or intermittent PrEP. This modality involves taking 2 oral pills 2–24 hours prior to sex, 1 oral pill 24 hours after the first dose, and 1 oral pill

24 hours after the second dose. There is scientific evidence that shows this modality provides effective HIV prevention for SMM engaging in condomless anal sex [10–12]. In December 2021, the U.S. FDA approved long acting cabotegravir (Apretude) as PrEP for use by adults and adolescents, who have increased vulnerability to HIV acquisition. Currently, it is designed to be administered through intramuscular injection every two months. A clinical trial of SMM, transgender women and cisgender women found cabotegravir to be superior to prevention HIV infection compared to daily oral PrEP [13]. The recent availability of multiple modalities provides a unique opportunity to increase uptake, especially among Black SMM.

Previous studies have found both on-demand and LAI-PrEP to be generally acceptable among SMM [14–18]. Studies assessing preference for PrEP modalities among SMM have yielded mixed results, with some showing daily oral [19–21] and others finding LAI-PrEP [17,18,22,23] to be the most preferred. However, a vast majority of these studies are quantitative, and few explore the reasons for those PrEP modality preferences. There is a dearth in the published literature on the acceptability of and preference for on-demand and LAI-PrEP among PrEP-eligible Black SMM. This study aimed to explore preference for on-demand and LAI-PrEP and reasons for those preferences among HIV-negative Black SMM of different PrEP use profiles (current PrEP users, current non-PrEP users, and PrEP discontinuers).

## Methods

### Sample

Between March 2022 and April 2023, we conducted 17 focus group discussions (FGDs) with a total of 58 HIV-negative Black SMM residing in the Washington metropolitan area. The FGDS were disaggregated byPrEP use profile: current PrEP users (n = 7), current non-PrEP users (n = 7), and PrEP discontinuers (n = 3). The FGDs broadly explored motivators for PrEP initiation and adherence, social network composition and influence on PrEP use or lack thereof, and feedback on a planned social network strategy intervention to increase PrEP initiation and adherence among Black SMM. The analyses for the current study will focus on responses to questions related to on-demand and LAI-PrEP. The FGDs were followed by a brief demographic questionnaire. Eligibility criteria included: 1) being 18 years or older, 2) assigned male sex at birth, 3) identify as African American/Black, 4) HIV-negative status, and 5) current residence in the Washington DC metropolitan area. Participants were recruited through flyers and posters to attract clients receiving services at the study site (Us Helping Us, People into Living Inc, Washington DC, outreach events at gay nightclubs, bars, and community events, 5) and word of mouth referrals.

### Study procedures

While most FGDs were conducted in person, due to the COVID-19 pandemic, some were conducted virtually. First, we conducted a rapid HIV test to confirm HIV negative status for all potential participants. Next, we reviewed the study goals and objectives, potential harms and benefits due to study participation, institutional review board contact information, and we then sought consent to enroll in the study. All study participants provided written informed consent prior to conducting any study procedures. All participants were assigned unique identifier (UID) numbers, and no identifying information was collected to maintain confidentiality. Participants were instructed to verbally state their UID each time they contributed to the conversation, to facilitate attribution of quotes during manuscript development. Interviews were conducted entirely in English by the authors (AO,AE, RE, & JS), were digitally recorded, and ranged in length from 1–1.5 hours. The interviewers are masters and doctoral-level researchers (3 males, 1 female) with over 30 years combined experience working with the target population. Each FGD had between 3–6 individual participants. The FGD interview guides have been included as supplementary files (S1 Checklist). Upon completion of the FGD, participants who were not on PrEP were provided more information about PrEP and linked to the PrEP coordinators at the study site to schedule a PrEP consultation. All study procedures were approved by the Pearl institutional review boards. Upon completion of the interview, participants were compensated $50 for their time.

## Data analysis

We utilized qualitative content analysis to analyze the transcribed FGDs. First, we reviewed all study transcripts and independently coded data into different NVivo nodes. Data were coded using "open coding", a process in which additional codes were made as needed throughout the process of reviewing transcripts. After the FGDs were coded, relevant quotes were further refined and organized into broad categories. For these analyses, we identified themes for the participant responses related to on-demand PrEP and LAI-PrEP.

## Results

### Participant demographics

Participant demographic as presented in the Table 1. A majority of the sample (57%) were between the ages of 25 and 34 years. Most identified as cisgender men (90%) and gay/homosexual (69%). Almost half (47%) were current PrEP users, 28% were PrEP discontinuers, and 26% were current non-PrEP users. While 98% identified as Black/African American, 11% considered themselves to be of Latino/Hispanic origin.

### Qualitative findings

Two key themes emerged around interest and preferences for on-demand PrEP and LAI-PrEP: 1) lack of interest in on-demand PrEP, and 2) high acceptability of LAI-PrEP. Additionally, there were some differing opinions on both PrEP modalities depending on participant current PrEP use status (current PrEP users, current non-PrEP users, and PrEP discontinuers).

### Theme 1: Lack of interest in on-demand PrEP

Most participants (76% of all FGDs) expressed a lack of interest in on-demand PrEP for various reasons. These sentiments were expressed after the moderators provided a brief overview of on-demand PrEP and the dosing strategy currently recommended by the CDC. The reasons for lack of interest were: 1) inability to accurately anticipate and plan for

**Table 1. Participant demographics.**

|  | Overall Sample (N = 58) |
| --- | --- |
| **Age** | |
| 18–24 years | 20 (34.5%) |
| 25–34 years | 33 (56.9%) |
| 35 years+ | 5 (8.6%) |
| **Race/Ethnicity** | |
| Black/African American | 57 (98.3%) |
| Other | 1 (1.7%) |
| **Gender** | |
| Cisgender male | 52 (89.7%) |
| Other | 6 (10.3%) |
| **Sexual Orientation** | |
| Gay/Homosexual | 40 (68.9%) |
| Bisexual | 10 (17.2%) |
| Other | 8 (13.8%) |
| **PrEP Status** | |
| Current PrEP users | 27 (46.6%) |
| PrEP discontinuers | 16 (27.6%) |
| Current non-PrEP users | 15 (25.9%) |

sexual activity in advance, 2) uncertainty about effectiveness of on-demand PrEP, and 3) potential for unnecessary medication use, especially when anticipated sexual activity doesn't occur.

Several participants described the spontaneous nature of sex and raised concerns about their ability to accurately anticipate when sexual activity will take place in concert with the 2-1-1 on-demand PrEP pill regimen. One participant who was currently using PrEP said:

> "That means you would literally have to know two days in advance that I'm going to have sex [...] it's not planned out like that. But it's kind of crazy that you have to actually plan." (40-45 years old, Gay, Current PrEP user Group)

Another participant, who was currently not using PrEP, mentioned that while sex can be anticipated to occur on special occasions such as Valentine's Day, it is generally spontaneous and unplanned:

> "I think that's a little complicated because you don't know when you want to have sex. You might run upon something and not have time, unless you're planning for Valentine's day, 'okay, I'm going to have sex,' then otherwise, no." (55-60 years old, Straight Current non-PrEP user Group)

Another reason for general lack of interest in on-demand PrEP were skepticism about the effectiveness of on-demand PrEP to prevent HIV infection, especially compared to daily oral PrEP. A participant, who is a current PrEP user, described having question about the level of protection against HIV provided by the on-demand PrEP pill regimen and further concerns about the number of pills needed to be taken and the timing of them:

> "I think also it's the full protection versus 'kind of' protection. When you have the PrEP [daily regimen] in your system, you can miss 1-2 days, and you'll still have it in your system, and you won't be nervous. With the on-demand, 'Did you take the one pill, or did you take the two days before?' Man, it's too much. It's too much." (40-45 years old, Gay, Current PrEP user Group)

Lastly, participants expressed concerns about the redundancy and potential harms of taking a medication they might not need, especially when anticipated sexual activity does not materialize into actual sex:

> "I don't want to put anything into my body that is not a guarantee of something. I don't want to premeditate that I'm going to have sex on a certain day because you don't know what's going to happen, you don't know how you're going to feel X, Y, and Z, that's number one." (20-25 years old, Gay, Current non-PrEP user Group)

> "The starting and stopping of it and what effect that has on the body and then also on the effectiveness of the drug. I'm not going to do that to myself. Yeah, I'm curious about that." (25-30 years old, Queer, Discontinued PrEP Group)

Theme 2: High acceptability of LAI-PrEP

Most participants (82% of all FGDs) found LAI-PrEP to be highly acceptable and this can be attributable to: 1) LAI-PrEP being convenient, and 2) LAI-PrEP being a potential solution to suboptimal adherence to daily oral PrEP due to forgetfulness. However, there were still concerns around LAI-PrEP specifically around the fear of needles and novelty of this new PrEP modality.

Several participants expressed prior awareness of LAI-PrEP and found it to be acceptable. The convenience of use, especially for individuals who were already on daily oral PrEP, was a major influencing factor for interest in LAI-PrEP:

> "I think it [LAI-PrEP]is easier. I travel a lot so trying to time my traveling around when the bottle will come and having to rearrange things. I do my testing regularly but trying to schedule it around that or just living in general. The whole pill

situation can be a lot depending on how often or not often they do your shipment." (30-35 years old, Bisexual, Current PrEP user Group)

**Another participant, who had discontinued PrEP, echoed these sentiments**

"I am very interested in injectable PrEP. I would prefer injectable PrEP over a pill for a multitude of reasons. It's easier, quicker. I don't have to worry about it." (55-60 years old, Gay, Discontinued PrEP Group).

The other common reason for the widespread acceptability of LAI-PrEP was the ability to circumvent the barriers associated with taking a daily oral pill. Across all three PrEP use profile, participants agreed that LAI-PrEP would be a viable PrEP option for them especially because it involves not having to remember to take a daily oral pill:

"If I was going to use PrEP, I would go that route [LAI-PrEP] because I have the worst memory in the world, so I would not remember to take the pills daily or after. I would just do one injection every two months." (35-40 years old, Gay, Current non-PrEP user Group)

"For discipline purposes, I would be interested in it [LAI-PrEP]. I don't have to worry about every morning getting the pill and taking it." (25-30 years old, Queer, Discontinued PrEP Group)

"But some people who are not used to taking medications and stuff, taking a pill orally may be hard for them, so then the injection may be better when you can make an appointment every two months, get a quick injection, and then you're set for the next two months." (40-45 years old, Gay, Current PrEP user Group)

Nonetheless, there remained reservations and concerns about LAI-PrEP across all PrEP use profile. Some participants expressed concerns about needles and that being a major barrier to them adopting LAI-PrEP:

"I'm sick of all these needles going into my body. I'm already vaccinated [for COVID] and they told me at first I was going to need one shot for that, and next thing I knew I needed another one. Then there's the booster and that whole thing. I don't like needles. (25–30 years old, Gay, Current non-PrEP user Group)

"I would not be open to LAI-PrEP because I hate being stuck, and I'm used to taking the pills. That's kind of like vitamins. I don't want to be stuck." (30-35 years old, Queer, Current PrEP user Group)

Others mentioned that the relative novelty of this new PrEP was a cause for them to not be an early adopter of LAI- PrEP:

"I have reservations regarding the newness of the injection. Also, taking a pill every day has become normalized so it would feel weird. I would be anxious because it would feel like I'm not doing something to protect myself." (30-35 years old, Gay, current PrEP user Group)

One participant, who had previously been on PrEP, specifically mentioned medical mistrust as barrier to PrEP adoption and the need to get information from a trusted source to feel comfortable adopting LAI-PrEP:

"I still think there is a lot of mistrust on this, on PrEP. Yeah, I think there's still a lot of mistrust and all these different options, I think can be overwhelming, especially if the information is not provided correctly. I need to hear from somebody I trust." (55-60 years old, Gay, Discontinued PrEP Group)

While there was consensus around LAI PrEP being acceptable and lack of interest in on-demand PrEP, there were some interesting differences by PrEP modality group. Firstly, individual who were currently on daily oral PrEP were more hesitant to alternative PrEP modalities, compared to individuals who were current non-PrEP users and individuals who had discontinued PrEP. Daily oral PrEP users expressed contentment with their current PrEP regimen and believed that adopting LAI-PrEP or on-demand PrEP would require more effort:

> "I would say my hesitation would be the constant scheduling. I'm not sure I would be on top of it. I could easily see myself canceling an appointment if I got busy or whatever. Whereas in the pill form, you're fully in control or whether you're taking it." (30-35 years old, Gay, current PrEP user Group)

> "When I first heard about it, I tried it because I was like, "You know what? This is a nice way to save on these pills [laughter], and I don't have to go and keep refilling them." And then one day I forgot, and I was like, "This was the day I was supposed to take them [laughter], man. Oh, no." So ever since that, I said, "I'll take it daily [laughter]. You can't predict the future." (40-45 years old, Gay, current PrEP user Group)

Secondly, current non-PrEP users and discontinued groups expressed that LAI-PrEP seemed more feasible than other PrEP modalities, especially daily oral:

> "For discipline purposes, I would be interested in LAI-PrEP. I don't have to worry about every morning getting the pill and taking it. I have to talk to my PCP [healthcare provider] ito see how this could affect my health, but other than that, I would be interested." (25-30 years old, Queer, Discontinued PrEP Group)

> "This is probably the most appealing PrEP sounded to me because I don't have to worry about taking it every day. I have to give myself insulin injections every day, numerous times, so needles aren't really a problem for me, but then, you know, I just have the appointment and then I'm on PrEP. That's the most appealing PrEP methodology for me." (25-30 years old, Bisexual, Current non-PrEP user Group).

## Discussion

The current study expands the growing literature on PrEP modality preferences among sexual minority communities. Our findings suggest limited interest in on-demand PrEP and high acceptability of LAI-PrEP among a sample of PrEP-eligible Black SMM. Additionally, participations expressed various concerns related to utilizing LAI-PrEP including fear of needles and medical mistrust. Lastly, current PrEP users were less receptive to alternative PrEP modalities and individuals who were current non-PrEP users and who had discontinued PrEP were more interested in LAI-PrEP. Our findings are paramount to informing current efforts to increase PrEP uptake and adherence among Black SMM, a group disproportionately affected by HIV and with suboptimal PrEP adoption rates.

Overall, most participants expressed a lack of interest in on-demand PrEP. This finding is consistent with other studies that have found lower usage of on-demand PrEP, compared to daily oral PrEP use, among SMM [24–26]. The reasons given for lack of interest were: 1) inability to accurately anticipate and plan for sexual activity in advance, 2) uncertainty about effectiveness of on-demand PrEP, and 3) potential for unnecessary medication use, especially when anticipated sexual activity doesn't occur. While daily oral PrEP is the most utilized PrEP modality, some SMM—who might not be acceptable to oral PrEP—are capable of accurately identifying specific periods and circumstances that increases chances for HIV infection. For SMM this can include participation in sex parties, annual pride celebrations, circuit parties, etc. Consequently, it is important that HCP present on-demand PrEP as an option to individuals who fall into these categories.

There were some expressed concerns about the effectiveness of on-demand PrEP in protecting against HIV infection. While on-demand PrEP has been proven to adequately protect against HIV when taken as prescribed [10–12], a

multi country study of SMM found higher level of HIV seroconversion among participants who utilized on-demand PrEP compared to those who utilized daily oral PrEP, with inadequate medication adherence being found in all individuals who seroconverted [27]. Consequently, it is imperative that clear and detailed instructions for the usage of on-demand PrEP are provided by the HCP to prevent HIV infection due to incorrect use.

Most participants found LAI-PrEP to be highly acceptable, which is consistent with findings of previous empirical studies among SMM [14–18]. The viability of LAI-PrEP hinged on its' convenience and ease of use, compared to daily oral PrEP. While LAI-PrEP is a viable and highly acceptable option for individuals predisposed to HIV infection, there still exists a myriad of barriers to uptake, especially among Black SMM. A recently published article of LAI-PrEP users and HCP found various barriers to LAI-PrEP uptake including lack of HCP awareness and knowledge, lack of insurance coverage, injection site reaction, fear of needles, lack of streamlined workflow and limited staff capacity [28]. These findings suggest that the mere approval and availability of LAI-PrEP might not result in widespread uptake and adherence. Programs to increase awareness and knowledge of LAI-PrEP among Black SMM and HCP that serve them are needed. Additionally, it is imperative that HCP who provide PrEP services receive proper training on the administration of LAI-PrEP and that there is dedicated staff to help clients navigate the insurance coverage process.

This study has several limitations. The study measures relied on participant recall/self-report, which may have contributed to social desirability bias. Also, we asked hypothetical questions about PrEP modality preferences, without providing details about possible side effects, costs, medical follow-up obligations, possible risks, etc. Further information about each PrEP modality can drastically change participants responses.

## Conclusion

The current study expands the growing literature on PrEP modality preferences especially among Black SMM in the U.S. Our findings suggest limited interest in on-demand PrEP and high acceptability of LAI-PrEP. It is important that healthcare providers present on-demand PrEP as an option to individuals who fall into these categories. Programs to increase awareness and knowledge of LAI-PrEP among Black SMM and HCP that serve them are needed. Additionally, it is imperative that HCP who provide PrEP services receive proper training on the administration of LAI-PrEP and that there is dedicated staff to help clients navigate the insurance coverage process.

## Supporting information

**S1 Checklist.**
(PDF)

## Acknowledgments

We thank all the participants of the study for their participation.

## Author contributions

**Conceptualization:** Adedotun Ogunbajo, DeMarc Hickson.

**Data curation:** Adedotun Ogunbajo, Alexa Euceda.

**Formal analysis:** Adedotun Ogunbajo, Alexa Euceda.

**Funding acquisition:** Adedotun Ogunbajo, Temitope Oke, DeMarc Hickson.

**Investigation:** Adedotun Ogunbajo, Raven Ekundayo, Jamil Smith, DeMarc Hickson.

**Methodology:** DeMarc Hickson.

**Project administration:** Adedotun Ogunbajo, Alexa Euceda, Raven Ekundayo, Jamil Smith, DeMarc Hickson.

**Resources:** Adedotun Ogunbajo, DeMarc Hickson.

**Supervision:** Adedotun Ogunbajo, Alexa Euceda, Temitope Oke, DeMarc Hickson.

**Writing – original draft:** Adedotun Ogunbajo.

**Writing – review & editing:** Alexa Euceda, Raven Ekundayo, Jamil Smith, Temitope Oke, DeMarc Hickson.

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
