## [Decision Letter · Decision Letter 0]

26 Dec 2024

PONE-D-24-43968Preferences for on-demand/intermittent/event-driven and long-acting injectable (LAI) HIV pre-exposure prophylaxis (PrEP) among HIV-negative Black gay, bisexual, and other sexual minority men in the United States: A qualitative studyPLOS ONE

Dear Dr. Ogunbajo,

Thank you for submitting your manuscript to PLOS ONE. After careful consideration, we feel that it has merit but does not fully meet PLOS ONE’s publication criteria as it currently stands. Therefore, we invite you to submit a revised version of the manuscript that addresses the points raised during the review process.

The paper is nearly suitable for publication. However, please address all of the comments from the two reviewers in your revision, which would be required to meet the criteria for acceptance for this journal.

We look forward to receiving your revised manuscript.

Kind regards,

Douglas S. Krakower, MD

Academic Editor

PLOS ONE

Journal Requirements:

 When submitting your revision, we need you to address these additional requirements.  1. Please ensure that your manuscript meets PLOS ONE's style requirements, including those for file naming. The PLOS ONE style templates can be found at  https://journals.plos.org/plosone/s/file?id=wjVg/PLOSOne_formatting_sample_main_body.pdf and  https://journals.plos.org/plosone/s/file?id=ba62/PLOSOne_formatting_sample_title_authors_affiliations.pdf. 

 2. Please amend either the title on the online submission form (via Edit Submission) or the title in the manuscript so that they are identical.  3. Please amend your list of authors on the manuscript to ensure that each author is linked to an affiliation. Authors’ affiliations should reflect the institution where the work was done (if authors moved subsequently, you can also list the new affiliation stating “current affiliation:….” as necessary). 4. Please match your authorship list in your manuscript file and in the system.   5. PLOS requires an ORCID iD for the corresponding author in Editorial Manager on papers submitted after December 6th, 2016. Please ensure that you have an ORCID iD and that it is validated in Editorial Manager. To do this, go to ‘Update my Information’ (in the upper left-hand corner of the main menu), and click on the Fetch/Validate link next to the ORCID field. This will take you to the ORCID site and allow you to create a new iD or authenticate a pre-existing iD in Editorial Manager.  6. We note that the grant information you provided in the ‘Funding Information’ and ‘Financial Disclosure’ sections do not match.   When you resubmit, please ensure that you provide the correct grant numbers for the awards you received for your study in the ‘Funding Information’ section. 7. Thank you for stating the following financial disclosure:   [This study was funded by the National Institutes of Health (R34DA054870). Dr. Ogunbajo acknowledges salary support from U.S. National Institutes of Health K01MH129165.].   Please state what role the funders took in the study.  If the funders had no role, please state: ""The funders had no role in study design, data collection and analysis, decision to publish, or preparation of the manuscript.""  If this statement is not correct you must amend it as needed.  Please include this amended Role of Funder statement in your cover letter; we will change the online submission form on your behalf.  8. We note that you have indicated that there are restrictions to data sharing for this study. PLOS only allows data to be available upon request if there are legal or ethical restrictions on sharing data publicly. For more information on unacceptable data access restrictions, please see http://journals.plos.org/plosone/s/data-availability#loc-unacceptable-data-access-restrictions.    Before we proceed with your manuscript, please address the following prompts:   a) If there are ethical or legal restrictions on sharing a de-identified data set, please explain them in detail (e.g., data contain potentially identifying or sensitive patient information, data are owned by a third-party organization, etc.) and who has imposed them (e.g., a Research Ethics Committee or Institutional Review Board, etc.). Please also provide contact information for a data access committee, ethics committee, or other institutional body to which data requests may be sent.  b) If there are no restrictions, please upload the minimal anonymized data set necessary to replicate your study findings to a stable, public repository and provide us with the relevant URLs, DOIs, or accession numbers. For a list of recommended repositories, please see https://journals.plos.org/plosone/s/recommended-repositories. You also have the option of uploading the data as Supporting Information files, but we would recommend depositing data directly to a data repository if possible.  We will update your Data Availability statement on your behalf to reflect the information you provide. 

Reviewers' comments:

Reviewer's Responses to Questions

**Comments to the Author**

1. Is the manuscript technically sound, and do the data support the conclusions?

Reviewer #1: Yes

Reviewer #2: Partly

2. Has the statistical analysis been performed appropriately and rigorously? 

Reviewer #1: N/A

Reviewer #2: N/A

3. Have the authors made all data underlying the findings in their manuscript fully available?

Reviewer #1: Yes

Reviewer #2: No

4. Is the manuscript presented in an intelligible fashion and written in standard English?

Reviewer #1: Yes

Reviewer #2: Yes

5. Review Comments to the Author

Reviewer #1: This is a well-written and timely paper. I have a few minor suggestions to improve the paper. Please clarify the following points:

- It appears that each focus group included people who were on-PrEP, PrEP-naive, and who had discontinued PrEP. How could these different groups have influenced or biased each other in their responses during the focus group? Was this controlled for in any way?

- A more detailed description of the qualitative data analysis is warranted. How many coders were there? How was interrater reliability assessed? As many items as you can complete from the COREQ (https://www.hsph.harvard.edu/wp-content/uploads/sites/2448/2021/02/Consolidated-criteria-for-reporting-qualitative-research-COREQ.pdf) would be informative.

- Was any other data gathered to assess participants' risk behaviors and how/whether they varied? Or the reasons why some of them had discontinued or decided not to get on PrEP? These could be informative contexts for their preferences around PrEP.

Reviewer #2: This is an interesting study with important implications for the development of HIV prevention interventions. I am particularly impressed with the researchers’ ability to recruit current PrEP users, non-users, and discontinuers, and discuss differences in PrEP attitudes between these groups. However, this paper would benefit from reframing and significant reorganization. Specific comments and suggestions are provided below.

1. The authors present this as a study of Black gay, bisexual, and other sexual minority men. However, their eligibility criteria referred to sex assigned at birth—not gender—and not all participants identified as cisgender men. Additionally, the eligibility criteria made no mention of sexual orientation, and only 69% of participants identified as gay/homosexual. It is unclear from the Results section how the other 31% of participants identified (i.e., queer, bisexual/pansexual, straight/heterosexual, etc.), but at least one participant is described as “straight,” suggesting that not all participants identified as men who have sex with men. Furthermore, although the eligibility criteria for the study included identifying as African American/Black, 2% of participants in the sample did not identify as Black. The researchers should provide a demographics table that clearly describes the gender, sexual orientation, and race/ethnicity of the participants in the sample, and should not describe the sample as “58 HIV-negative Black SMM” if this does not accurately reflect the identities of all participants.

2. Relatedly, it is commendable that the study sample included current PrEP users, PrEP non-users, and PrEP discontinuers, but it would be helpful if a demographics table noted how many participants fell into each category. Additionally, the authors seem to use the phrases “non-PrEP users” and “PrEP Naive Group” interchangeably, but these are not synonymous; one describes participants’ lack of experience taking PrEP, while the other describes participants’ lack of knowledge of PrEP. If all PrEP non-users in the sample were also PrEP naïve, this should be specified within the manuscript.

3. The authors make a strong argument for focusing on Black SMM (because they are disproportionately affected by HIV, and less likely than their white counterparts to receive PrEP prescriptions). Although the introduction references structural inequities that contribute to these health disparities (e.g., racism), the authors may wish to expand their discussion of these systemic/structural factors, and emphasize them over patient-level factors (e.g., likelihood of following providers’ recommendations) that may be seen as blaming patients. Additionally, I would advise the authors to move away from "risk" language throughout the paper, as it can be stigmatizing.

4. The authors organized the findings into two themes, which they defined as “lack of interest in on-demand PrEP” and “high acceptability of LAI-PrEP.” They then identify numerous “reasons” or “common sentiments” expressed by participants. The Results section would benefit from a reorganization that presents these common sentiments as the themes of interest, or as sub-themes under the previously identified umbrella categories. It would also be useful to provide more specific information about the number of participants who expressed each sentiment.

5. According to the Methods section, the focus groups explored “PrEP initiation and adherence, social network composition and influence on PrEP use or lack thereof, and feedback on a planned social network strategy intervention to increase PrEP initiation and adherence among Black SMM.” It is unclear why analyses then exclusively focus on attitudes towards on-demand and LAI-PrEP. Specifically, why did the researchers not discuss attitudes towards daily dosing? Given that daily dosing was mentioned by multiple participants and is the only FDA-approved PrEP schedule, the omission was confusing.

6. The authors provide very interesting illustrative quotations throughout the manuscript, but do not always integrate the quotations into the surrounding text. They may consider including fewer examples within the body of the text, and referencing a table with additional illustrative quotations.

6. PLOS authors have the option to publish the peer review history of their article (what does this mean? ). If published, this will include your full peer review and any attached files.

**Do you want your identity to be public for this peer review?** For information about this choice, including consent withdrawal, please see our Privacy Policy .

Reviewer #1: **Yes: ** Rebecca Giguere

Reviewer #2: No

---

## [Editor Report · Decision Letter 1]

4 Apr 2025

Preferences for on-demand/intermittent/event-driven and long-acting injectable (LAI) HIV pre-exposure prophylaxis (PrEP) among HIV-negative Black gay, bisexual, and other sexual minority men in the United States: A qualitative study

PONE-D-24-43968R1

Dear Dr. Ogunbajo,

We’re pleased to inform you that your manuscript has been judged scientifically suitable for publication and will be formally accepted for publication once it meets all outstanding technical requirements.

Kind regards,

Douglas S. Krakower, MD

Academic Editor

PLOS ONE
---

## [Editor Report · Acceptance letter]

PONE-D-24-43968R1

PLOS ONE

Dear Dr. Ogunbajo,

I'm pleased to inform you that your manuscript has been deemed suitable for publication in PLOS ONE. Congratulations! Your manuscript is now being handed over to our production team.

Kind regards,

on behalf of

Dr. Douglas S. Krakower

Academic Editor

PLOS ONE